# Meta-QTL and Candidate Gene Analyses of Agronomic Salt Tolerance and Related Traits in an RIL Population Derived from *Solanum pimpinellifolium*

**DOI:** 10.3390/ijms25116055

**Published:** 2024-05-31

**Authors:** Maria J. Asins, Emilio A. Carbonell

**Affiliations:** Instituto Valenciano de Investigaciones Agrarias (IVIA), 46113 Moncada, Valencia, Spain

**Keywords:** tomato, fruit yield, water content, aquaporins, nutrients, rootstock breeding, wild germplasm

## Abstract

Breeding salt-tolerant crops is necessary to reduce food insecurity. Prebreeding populations are fundamental for uncovering tolerance alleles from wild germplasm. To obtain a physiological interpretation of the agronomic salt tolerance and better criteria to identify candidate genes, quantitative trait loci (QTLs) governing productivity-related traits in a population of recombinant inbred lines (RIL) derived from *S. pimpinellifolium* were reanalyzed using an SNP-saturated linkage map and clustered using QTL meta-analysis to synthesize QTL information. A total of 60 out of 85 QTLs were grouped into 12 productivity MQTLs. Ten of them were found to overlap with other tomato yield QTLs that were found using various mapping populations and cultivation conditions. The MQTL compositions showed that fruit yield was genetically associated with leaf water content. Additionally, leaf Cl^−^ and K^+^ contents were related to tomato productivity under control and salinity conditions, respectively. More than one functional candidate was frequently found, explaining most productivity MQTLs, indicating that the co-regulation of more than one gene within those MQTLs might explain the clustering of agronomic and physiological QTLs. Moreover, MQTL1.2, MQTL3 and MQTL6 point to the root as the main organ involved in increasing productivity under salinity through the wild allele, suggesting that adequate rootstock/scion combinations could have a clear agronomic advantage under salinity.

## 1. Introduction

Crop breeding and land management strategies are key to maximizing the utilization of finite resources to mitigate climate change and ensure global food availability [1]. For this purpose, the adaptative genetic variants (alleles) responsible for drought, flood, extreme temperature, and salinity tolerance must be discovered, often within wild-crop relatives because these alleles were mostly lost during domestication and breeding [2,3]. Thus, properly documented germplasm collections including crop wild relatives and segregating populations obtained by crossing cultivated and wild species (prebreeding populations) are fundamental materials for descrying tolerance alleles [4].

Salinity tolerance is, among abiotic stresses, the most complex trait to improve in crops since it depends on several factors such as the type of salinity, plant developmental stage, and stress intensity, which may all vary considerably in the field [5]. Saline soils can be managed with the use of high-quality water, but water availability for agriculture is decreasing or becoming erratic due to climate change. Thus, the use of poor-quality water irrigation is becoming frequent in arid and semi-arid regions, expanding and/or increasing soil salinization, which makes breeding for salt-tolerant crops necessary to reduce food insecurity.

Tomato (*Solanum lycopersicum* L.) is the most important fruit crop in the world (above bananas, watermelons, apples, grapes and oranges) and the second most important horticultural crop after potato [6], mostly due to the versatility of its fruit and its nutritional value [7]. In fact, tomato fruit is an important component of daily meals in many countries and constitutes a major source of minerals, vitamins, and antioxidant compounds. Fortunately, a huge body of genetic, physiological, molecular, and genomic knowledge has been accumulated regarding this species that has facilitated the quick development of biotechnological tools [8,9,10,11,12,13] to support research further and speed up tomato breeding programs. Research on the diversity of its wild genetic resources is abundant, particularly regarding salt tolerance [7,13,14,15,16]. From the literature, it can be concluded that (1) there is genetic variability to improve tomato salt tolerance and (2) the salt tolerance response of accessions belonging to different wild species may lie upon different genes that we must bring to light.

Salt tolerance mechanisms were initially classified into three categories [17]: (1) tolerance to osmotic stress that reduces cell expansion in root tips and young leaves and causes stomatal closure, (2) Na^+^ exclusion from leaves, and (3) tissular tolerance to accumulate Na^+^ (or Cl^−^). Importantly, the plant organ contribution of each mechanism to the plant salt tolerance may be different (i.e., sink versus source organs, root versus above ground parts of the plant). In this sense, the expression level of genes involved in Na^+^ extrusion, Na^+^ compartmentation, K^+^ nutrition, cell expansion and division, osmotic adjustment, reactive oxygen species scavenging, and chemical and phytohormone signaling (Ca^2+^, abscisic acid, auxin, etc.), relevant for salt tolerance, depends not only on the wild accession but also on the plant organ [18,19]. Moreover, in the case of crop plants, it is ultimately the yield under salinity that determines whether a gene (or a given set of linked genes) is of agronomic importance.

Given such complexity, how can we efficiently uncover salt tolerance alleles in the tomato wild germplasm? Quantitative trait locus (QTL) analysis of salt tolerance in terms of fruit yield using fixed segregating populations derived from wild tolerant accessions (mostly from *S. pimpinellifolium, S. galapagense* and *S. pennellii*) have been useful for locating the genomic regions involved [15,20]. However, the fruit size and yield of these materials are always small in comparison with those of commercial tomato cultivars, which makes the interpretation of the agronomic salt tolerance phenotypes tricky. The development of marker technology has allowed both the saturation of genetic maps and their anchorage to the tomato physical map, which makes candidate gene analysis within QTL regions feasible. Nowadays, several genome-wide association studies (GWAS) using agronomically characterized tomato germplasm collections [21,22,23,24] have, theoretically, increased the resolution of QTL mapping relative to bi-parental populations. However, the evaluation of agronomic salt tolerance in wild germplasm seems too complex for this QTL methodology [3,16]. A strategy to simplify the evaluation of agronomic salt tolerance is testing just the root effects of the segregating–prebreeding population on the fruit yield of a commercial variety under salinity by using that population as rootstocks [25,26].

Given that some of our previous studies (from 2001 to 2010) on agronomic salt tolerance and related physiological components, such as leaf ion (Na^+^, K^+^, Ca^2+^, Cl^−^) concentration, water content, and reproductive traits, were carried out using the same RIL population E9xL5 derived from *S. pimpinellifolium* [20,25,27,28], whose linkage map was fully developed afterwards [26], a major objective of the present study is the QTL reanalysis of those evaluated traits using this SNP-saturated linkage map. Secondly, to obtain both a physiological interpretation of the agronomic salt tolerance and better criteria to identify candidate genes, the obtained QTLs will be clustered using QTL meta-analysis to synthesize QTL information from the diverse experiments and narrow down chromosomal regions that control trait variation [29]. Finally, we will explain ways to exploit the genetic knowledge provided in obtaining salinity-tolerant tomato plants.

## 2. Results

The list of the 85 significant QTLs detected with the SNP linkage map for traits evaluated in previously reported experiments is shown in Appendix A. Sixteen previously unreported QTLs correspond to reproductive traits. A total of 35 out of 69 QTLs (around 50%) were not detected in previous studies as a consequence of the linkage map used. The allele increasing the trait mean came from *S. pimpinellifolium* in 51 out of the 85 QTLs (60%). The projection on the SNP linkage map of the 85 QTLs governing the four types of traits (yield-, reproduction-, ion- and water-related traits) and those reported by Asins et al. [26] is shown in Figure 1.

This projection was used for metanalysis to classify all QTLs into clusters. Thus, 60 QTLs were grouped into 12 productivity MQTLs in chromosomes 1 (2), 2 (2), 3, 4 (2), 5, 6, 7 (2), and 11 (Table 1). All MQTLs, except for MQT7.1, which incorporates a QTL for the number of flowers per truss, include fruit yield QTLs. Contrary to expectations, yield QTLs detected using the RIL population did not align with the yield QTLs detected when the RIL population was grafted with a commercial variety.

As the Venn diagram shows in Figure 2, fruit yield (FY) QTLs, disregarding the salinity level, were most frequently grouped with water content (WC) QTLs (five MQTLs), followed by Na^+^ content QTLs (four MQTLs), QTLs for reproductive traits (R), Ca^2+^ and K^+^ contents (three MQTLs each), and Cl^−^ content QTLs (two MQTLs). When considering yield QTLs under salinity, the leaf K^+^ content becomes at least as important as the leaf water content in terms of the conjunction of their QTLs and yield QTLs in the composition of MQTLs depicted in Figure 2.

To explore the genes within each MQTL, their gene content (the genes within the MQTL confidence interval) was used for enrichment analysis. Some pathways were significant in seven MQTLs (Appendix A). Pathway RNA modification was found to be enriched in two of them (MQTL1.2 and MQTL3), and RNA processing was enriched in MQTL1.1. Positive regulation of transcription in MQTL4.2 and chromatin organization in MQTL5 were also observed. Amine transport and cellular response to nitrogen starvation presented the lowest false discovery rate (FDR) within MQTL2.2 and MQTL7.2, respectively. Additionally, to select the most likely candidates, previous information in the literature, the presence of sequence polymorphisms, and the level of transcription in relevant organs were investigated. Thus, a reduced list of putative candidates was obtained for each MQTL (Appendix A). Frequently, more than one known gene involved in variation for agronomic traits were found within the confidence intervals of MQTLs. Thus, MQTL1.2 includes auxin response factor 18, the abscisic receptor PYL8, several ferric reductase oxidases, the aquaporin PIP1.2, and the LBD18 coding genes; MQTL3 includes DXS, KLUH and expansin coding genes; MQTL4.1 includes NHX2 and CLV2 coding genes; MQTL4.2 includes cell growth defect factor 2 and auxin response factor 5; MQTL5 includes the cell division protein kinase 10 and the alpha-glucosidase I coding genes; MQTL11 includes genes coding for the auxin response factor 4, the aquaporin PIP2.6, and several nitrate transporters; MQTL2.2 includes the TM29 MAD-box transcription factor and the ORFX cell number regulator 1 coding genes; and MQTL6 includes the auxin efflux facilitator PIN6, the NAM1 transcription factor, the abscisic receptor PYR1, the nitrate transporter NPF6.3, a trehalose 6-phosphate phosphatase, and the aquaporin TIP2.3 coding genes.

## 3. Discussion

This study on tomato salt tolerance focusses on agronomic traits including not only the fruit yield but also reproductive traits such as the number of flowers, fruits per truss, and fruit set because they are related to crop productivity. Thus, MQTL housing reproductive trait QTLs frequently have fruit yield QTLs under control (MQTL1.1) and salinity conditions (MQTL4.2 and MQTL1.2) (Figure 2). Previously reported QTLs for reproductive traits [23,30,31] overlap with MQTL1.1, MQTL1.2, and MQTL4.2.

MQTL7.1 includes a QTL for the number of flowers per truss and QTLs for leaf Na^+^ and K^+^ contents whose underlying genes are *HKT1;2* and *HKT1;1* [32,33]. Although MQTL7.1 encloses no fruit yield QTL, by silencing *HKT1;2*, it was shown that this transporter reduces the flower Na^+^ content, alleviating the decline in tomato fruit yield under salinity [34].

### 3.1. Fruit Yield Is Frequently Associated with Leaf Water Content in the RIL Population

The preferential clustering of fruit yield QTLs observed (Figure 2) underscores the importance of some physiological traits contributing to salt tolerance. When plants grow under salinity conditions, a major and primary problem they have to face is loss of water due to decreased osmotic pressure [35]. Plants can compensate for this through various mechanisms such as (1) redirecting their energy resources to the synthesis and accumulation of organic solutes [36]; (2) the use of Na^+^ as an osmolyte, which requires tissue tolerance [37]; (3) regulating the amount and tissular distribution of aquaporins; (4) changing their endodermis and cell wall composition [38]; and (5) decreasing transpiration by stomata closure [7,35]. In the experiments under consideration with the RIL population derived from *S. pimpinellifolium*, the alleles increasing leaf water content and fruit yield under salinity are the same in all concerned MQTLs (MQTL3, MQTL5, and MQTL11 in Appendix A). Moreover, two of those MQTLs containing water content QTLs also include leaf Na^+^ content QTLs (MQTL3 and MQTL5), the increasing allele for both traits being the same in all of them, which suggests that Na^+^ could contribute to osmotic adjustment, as previously reported [39,40]. These results seem to point to the low-cost mechanisms (2), (3), and (4) as the main ones involved in compensating for decreased osmotic pressure due to salinity.

A higher capacity to maintain root hydraulic conductivity under salt stress in an accession of *S. pimpinellifolium* than in the cultivar “M82” was previously attributed to PIP1 transcript abundance [41]. Aquaporins, as functional candidates, were found in MQTL1.1 and MQTL11 (Solyc01g056720 and Solyc11g069430, respectively) and in other MQTLs housing yield QTLs but not water content QTLs: MQTL1.2 (Solyc01g094810), MQTL2.1 (Solyc02g071920), and MQTL6 (Solyc06g060760).

With regard to mechanism (4), change in the endodermis and cell wall composition, an expansin, (Solyc03g115890), and the alfa-glucosidase I (Solyc05g015250) coding genes could be involved in phenotypic differences at MQTL3 and MQTL5, respectively (Appendix A).

Therefore, these MQTL genes, putatively involved in minimizing salinity-induced decline in the leaf water status throughout the use of Na^+^ as an osmolyte, the regulation of aquaporins, and the change in endodermis and cell wall composition, could contribute to a higher fruit yield (mostly the fruit number component), making the tomato plant’s response to salinity tolerant from an agronomic point of view.

### 3.2. Leaf Nutrient Content and Tomato Productivity Are Connected in Most MQTLs

As judged by the MQTL composition (Figure 2) and the direction of gene effects (Appendix A), the fruit yield under the control condition is genetically and positively associated with the leaf Cl^−^ content. Thus, both the leaf Cl^−^ content and yield QTLs detected under control conditions are included in MQTL1.1 and MQTL2.2, supporting the relevance given to Cl^−^ as a nutrient [42,43,44,45]. A porin/voltage-dependent anion-selective channel protein and AE family transporter anion exchange coding genes (Solyc01g010760 and Solyc01g057770, respectively) are functional candidates included in MQTL1.1.

Four MQTLs containing productivity QTLs under salinity also have leaf K^+^ content QTLs (MQTL2.1, MQTL3, MQTL7.1, and MQTL11). From them, leaf K^+^ and Na^+^ content QTLs are included in MQTL7.1 and MQTL3, where the direction of gene effects is opposite at both. This clustering of Na^+^ and K^+^ QTLs might be related to the importance of maintaining an optimal K^+^/Na^+^ ratio in the cytoplasm under salinity, which has been pointed out as an essential mechanism of tolerance in plants [46,47,48]. Three functional candidates, involved in K^+^ transport, are found in MQTL2.1 (Solyc02g070290, Solyc02g071140, and Solyc02g070530).

Most fruit yield MQTLs include leaf nutrient content QTLs, supporting the idea that the uptake and distribution of nutrients (particularly K^+^) within the plant is affected by salinity and contribute to the final fruit yield of the tomato. In the case of MQTL6, where only fruit yield QTLs of the grafted population under salinity were included, it overlaps with nitrogen leaf content QTLs previously reported in the same grafted population [49], in agreement with the importance of N nutrition in yield, such as in other crops [50]. Nitrate transporters coding genes were found in MQTL6 (Solyc06g060620) and MQTL11 (Solyc11g069740, Solyc11g069710, and Solyc11g069760). Moreover, MQTL2.2 is significantly enriched in genes of the nitrate import and nitrate transport biological processes (Appendix A). Iron nutrition could also be important. Three genes related to iron uptake and homeostasis are located in MQTL1.2 (Solyc01g094900, Solyc01g094910, and Solyc01g094890), which overlaps with a QTL reported for leaf Fe content under low Fe nutrition in the same grafted population [51].

### 3.3. From Genetic Knowledge to Improving Salt Tolerance in the Field

Most fruit yield MQTLs reported here overlap with other tomato yield QTLs that were found using other mapping populations, cultivation conditions, and QTL detection methodologies such as GWAS. A summary of coincidences is shown in Table 2. Remarkable examples of frequent coincidences are MQTL2.2 [21,24,52,53,54], MQTL3 [23,54,55], MQTL4.1 [22,23,56], MQTL5 [23,24,56], and MQTL7.2 [56,57,58]. Such frequent coincidences seem to indicate that these MQTLs control tomato productivity across populations and cultivation conditions. The direction and strength of gene effects at these MQTLs vary according to the particular alleles at play and growing conditions.

Three genes underlying fruit weight QTLs have been cloned in tomato: *fw2.2*, *fw3.2*, and *fw11.3* [52,55,59], with a yet-undisclosed clear function for the respective encoded proteins [60]. *Fw2.2* (Solyc02g090730, a negative regulator of cell division from the cell number regulator CNR family, [52]) and *fw3.2* (Solyc03g114940, SlKLUH, a P450 enzyme of the CYP78A subfamily, [55]) are included in MQTL2.2 (containing fruit yield QTLs of the non-grafted population under both control and salinity conditions) and MQTL3 (containing fruit yield QTLs detected only in the grafted population under salinity), respectively. These results suggest that the relevant segregating genes involved in MQTL2.2 are crucial in the aerial part of the plant, but not so much in the root. On the contrary, there must be an important root component in the function of MQTL3 (containing *fw3.2* gene) under salinity that had not been previously observed. MQTL3 seems particularly relevant for rootstock breeding purposes because (1) it has a narrow confidence interval (Table 2) and (2) is involved in many physiological traits (Figure 2), (3) the increasing allele is generally the same (the wild allele), and (4) it contributes to plant salt tolerance.

**Table 2 ijms-25-06055-t002:** MQTLs including previously reported genes associated with fruit-yield-related traits.

MQTL	References				
1.1	Pons et al. [24]	Tsutsumi-Morita et al. [56]			
2.1	Tsutsumi-Morita et al. [56]				
2.2	Frary et al. [52]	Ampomah-Dwamena et al. [53]	Diouf et al. [54]	Mata-Nicolas et al. [21]	Pons et al. [24]
3	Chakrabarti et al. [55]	Diouf et al. [54]	Ye et al. [23]		
4.1	Ye et al. [23]	Kim et al. [22]	Tsutsumi-Morita et al. [56]		
4.2	Liu et al. 2018 [61]	Ye et al. [23]			
5	Ye et al. [23]	Tsutsumi-Morita et al. [56]	Pons et al. [24]		
6	Tsutsumi-Morita et al. [56]				
7.2	Nitsch et al. [57]	Tsutsumi-Morita et al. [56]	Torgeman and Zamir [58]		
11	Diouf et al. [54]				

Other relevant MQTLs are MQTL1.2, MQTL2.1, and MQTL6, which correspond to yield QTLs detected in the grafted population under salinity and include rootstock-mediated leaf nutrient (Fe, K, and N, respectively) content QTLs [49,51]. Moreover, two of them (MQTL1.2 and MQTL2.1) include known genes (*LBD18* and *PTI5*, respectively) involved in the development of the graft union in tomato [62]. Therefore, these three additional MQTLs appear relevant for rootstock breeding purposes to increase fruit yield under salinity.

Modern tomato commercial varieties are based on an almost exclusive *S. lycopersicum* (*L*) gene pool with very few wild genes, mostly related to disease resistance. The results here provided on fruit yield MQTLs show that some of them point to the root as the main organ involved under salinity, and the wild *S. pimpinellifolium* allele (*P*) as the increasing one (i.e., MQTL1.2, MQTL3, and MQTL6), which suggests that adequate rootstock/scion combinations (*PP*/*LL*) at them could be a new way to obtain a “heterotic” physiological phenotype of the plant with a clear advantage under salinity and, possibly other limiting environments (Figure 3). Thus, breeding materials obtained from the rich-well-adapted wild genetic resources could be more easily used as rootstocks of modern varieties to exploit the robustness (stability) of such combinations for tomato yield at least under salinity, and possibly other abiotic stresses where nutrient availability is limited.

In general, more than one functional candidate can be found, explaining most of the MQTLs (Appendix A). Thus, PIN6 Auxin transporter, involved in vascular tissue differentiation, root elongation, as well as growth responses to environmental stimuli [63], is located in MQTL6, in addition to other candidates (i.e., abscisic acid receptor PYR1, aquaporin TIP2.3, and nitrate transporter NPF6.3). Moreover, the auxin response factors Arf4 and Arf5, known to be related to salt tolerance [61,64,65,66], are within MQTL11 (also including aquaporin PIP2.6 and several nitrate transporters) and MQTL4.2 (also including cell growth defect factor 2), respectively. Therefore, an abundance of functional candidates within MQTLs seems to be the rule rather than the exception. This could indicate that the co-regulation of more than one gene within those MQTLs might explain the clustering of agronomic and physiological QTLs. In any case, gene knockouts produced by CRISPR/CAS9 can be used for phenotypic validation of numerous candidate genes covering the gap between this genetic knowledge obtained in prebreeding populations and crop improvement in the field [1].

## 4. Materials and Methods

### 4.1. Experiments, Traits, and QTL Analysis

Salt tolerance in terms of fruit yield was studied using a population of recombinant inbred lines (from F6 to F9 generation) derived from a single seed descendent from the hybrid between a salt-sensitive genotype of *Solanum lycopersicum* var. Cerasiforme (formerly L. esculentum) and a salt-tolerant line from *S. pimpinellifolium* L. (formerly *L. pimpinellifolium*) in different experiments from 2001 to 2010 [20,25,27,28]. These experiments, summarized in Appendix A, were performed using the grafted and non-grafted RIL populations, and several levels of salinity (electric conductivities of nutrient solutions), from 0.3 up to 15 dS/m. Moreover, experiments using the non-grafted population of RILs in Valencia (2001) and Malaga (2003) considered two salinity levels each. The control level was the same (electrical conductivity, E. C., of nutrient solution 0.3 dS/m) but differed for the salinity treatment, 15 and 9.5 dS/m, respectively. Salinity treatments in the experiments that used the population of RILs as a rootstock of commercial tomato cultivar Boludo were carried out in Murcia (2007) and Valencia (2013) and differed in salinity levels, which were 13.7 and 8.9 dS/m, respectively.

Traits were named as in the original references. When the traits were evaluated in the RIL population as rootstocks, a “g” was added before the trait code. Evaluated traits from the abovementioned experiments (Appendix A) were classified into four types: fruit yield traits (number of fruits, FN or gFN, total fruit weight, TFW or gTFW, and mean fruit weight, FW or gFW), reproductive traits (number of flowers per truss, FL, number of fruits per truss, FR, and fruit set percentage per truss, FS), leaf ionomic profile (Na^+^, K^+^, Ca^2+^, and Cl^−^), and water content traits (the water content, WC, in mg/g of dried weight of the two parts of the second and fifth leaves, the rachis and the leaflets, gRa2WC, gRa5WC, gLl2WC and gLl5WC, respectively). In the case of the last experiment [26], fruits were classified into small and commercial (large), depending on their weight. Fruits larger than 5 g were used to estimate commercial fruit yield under moderate salinity (gTFWl), and the rest, considered small fruits, were used to estimate gFNs and gTFWs.

Data from all these traits have been reanalyzed to detect QTLs using the SNP linkage map and methodology reported elsewhere [26]. This linkage map is based on 1899 non-redundant SolCAP SNPs, covering 1326.37 cM of genetic length. Genotypes from the recombinant population were obtained at F10 for 7720 SNPs from the SolCAP tomato panel (Illumina BeadXhip WG-401-1004). QTL analysis of traits was carried out using interval mapping (IM), and multiple-QTL mapping (MQM), which are procedures in MapQTL^®^ 6 [67]. A 5% experiment-wise significance level controlling for dependent markers was assessed using permutation tests. These LOD critical values ranged from 1.8 to 2.3 depending on the trait and chromosome. Significant QTLs were named as the trait followed by the chromosome where they were located and the suffix _C when detected under salinity conditions. QTLs for reproductive traits were not previously reported.

### 4.2. QTL Projection and Meta-QTL Analysis

A file (Appendix A) containing the genetic information of all detected QTLs was prepared according to the requirements of BioMercator v4.2.3 software [68]. QTL confidence intervals were calculated following Guo et al. [69]. QTL meta-analysis was conducted using the algorithm developed by Goffinet and Gerber [29] and the projected QTLs on the SNP-saturated linkage map previously reported [26]. Thus, the number of genomic regions containing QTLs (1, 2, 3, 4, or N) controlling a given trait or related ones from independent experiments is modeled. The most likely QTL arrangement is determined by means of the maximum likelihood method, and an Akaike-type statistical criterion indicates the best model among the five ones (that with the lowest Akaike). MQTLs containing QTLs for productivity-related traits were named based on the chromosome they are located at.

### 4.3. Identification of Candidate Genes

Left and right SNP markers of each MQTL confidence interval containing yield-related QTLs were identified using the linkage map and anchored to the Tomato SL2.50 ITAG2.4 using JBrowse 1.16.11 at the Sol Genomic Network [11] to download the included genes (mRNA). ShinyGO 0.76 [70] was used for the functional enrichment analysis of genes within each MQTL on default parameters (in all cases, FDR < 0.05).

Genes within each MQTL were also studied for the presence of frameshift InDels in the parental genomes (E9 and L5) [71]. Root expression of candidate genes (Illumina-drive and RPKM-normalized) was inferred from the Heinz cultivar using the tomato Plant eFP Browser (https://bar.utoronto.ca/eplant_tomato/ (accessed on 23 February 2024)), and from LA1589 (*S. pimpinellifollium*) using the Illumina RNAseq transcriptomic analysis (http://ted.bti.cornell.edu/cgi-bin/TFGD/digital/search.cgi?ID=D006 (accessed on 23 February 2024)). When an MQTL contained QTLs for fruit yield and/or reproductive traits detected in the non-grafted population of RILs, candidate gene expression at the fully opened flower was also considered.

## 5. Conclusions

In this paper, through Meta-QTL analysis, we investigated the clustering of QTLs controlling tomato salt tolerance in terms of fruit yield and three types of related traits: reproductive function and leaf water and ion (Na^+^, K^+^, Ca^2+^, Cl^−^) contents. Thus, twelve productivity MQTLs were obtained. Ten of them overlap other yield-related QTLs/genes detected using various populations, cultivation conditions, and statistical methods. Under salinity, fruit yield QTLs frequently grouped with QTLs controlling leaf water and K^+^ contents. Candidate gene analysis within confidence intervals of MQTLs showed that more than one functional candidate can be found, explaining most of the productivity MQTLs and suggesting that their co-regulation might be involved. Since some MQTLs (i.e., MQTL1.2, MQTL3, and MQTL6) point to the root as the main organ involved in productivity under salinity, a way to take advantage of their wild *S. pimpinellifolium* increasing allele (*P*) could be the selection of adequate rootstock/scion combinations (*PP*/*LL*) to obtain a “heterotic” physiological phenotype of the plant with a clear advantage under salinity and, possibly, other nutrient-limiting environments.

## Figures and Tables

**Figure 1 ijms-25-06055-f001:**
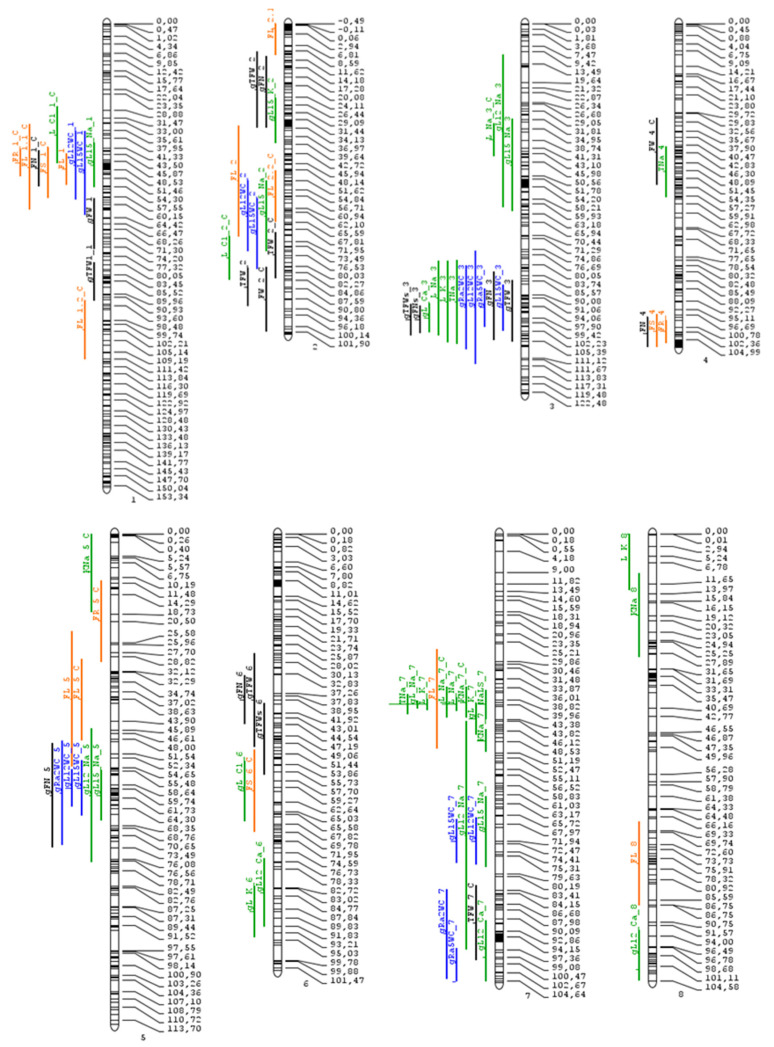
Distribution of QTLs (Appendix A and those reported [26]) throughout all 12 chromosomes of the genetic linkage map. Bars on the left side of the chromosomes correspond to confidence intervals of QTLs for traits related to fruit yield (black bars), water content (blue bars), nutrient content (green bars), and reproductive function (orange bars). Black bars within chromosomes represent marker density. The numbers on the right side correspond to the genetic distance in cM.

**Figure 2 ijms-25-06055-f002:**
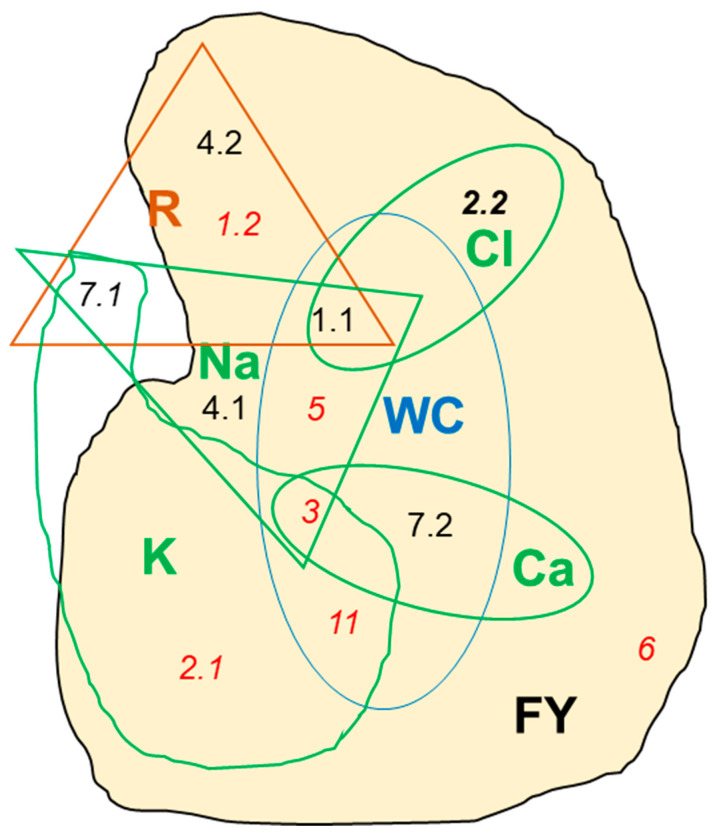
Venn diagram of MQTLs and type of affected trait: FY (fruit yield), WC (water content), Na^+^, K^+^, Ca^2+^ and Cl^−^ (nutrient contents) and R (reproductive). MQTLs including fruit yield QTLs have a yellow background. MQTLs in italics include salinity-detected QTLs; bold italics indicates QTLs detected under both conditions. MQTLs in red correspond to the grafted RIL population, and those in black correspond to the non-grafted population.

**Figure 3 ijms-25-06055-f003:**
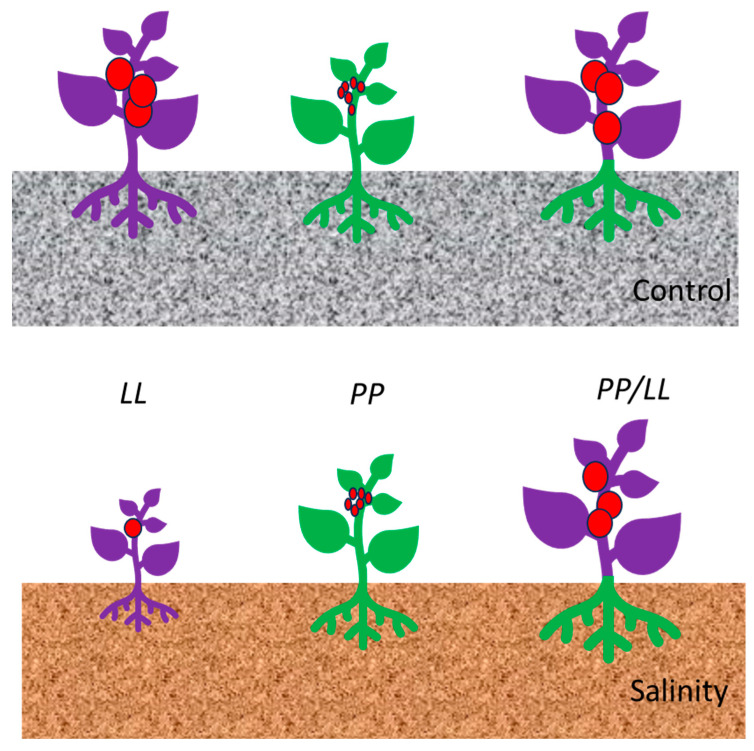
Graphical representation of the agronomic performance, under control (grey) and moderate salinity (brown) conditions, of non-grafted plants (commercial tomato cultivar in violet, and prebreeding salt-tolerant material containing *S. pimpinellifolium* alleles, *PP*, in green) and their salt-tolerant rootstock/scion combination (bicolor plants).

**Table 1 ijms-25-06055-t001:** List of productive MQTLs and their characteristics: mean position (Pos.) in cM, confidence intervals (CI) presented as the left and right (CI_lo and CI_hi) positions (in cM), and the nearest solcap_snp_sl markers (SNP). The physical length of the confidence interval (Mbp) and number of genes included (CI genes) are also indicated.

MQTL	Pos.	CI_lo (cM)	CI_lo SNP	CI_hi (cM)	CI_hi SNP	CI (Mbp)	CI Genes
1.1	47.03	44.6	33677	49.46	50504	71.06	1210
1.2	90.68	85.24	34545	96.12	28168	2.34	291
2.1	27.64	21.89	33633	33.38	29543	2.7	358
2.2	82.34	78.48	50066	86.2	67052	2.07	276
3	96.53	94.32	62180	98.74	61967	0.88	123
4.1	46.91	40.1	41577	53.73	24557	50.33	1096
4.2	102.54	99.77	47540	105.32	3896	0.87	113
5	60.13	57.24	50872	60.03	51127	4.05	209
6	41.38	37.01	55858	45.75	1360	1.95	198
7.1	40.04	39.4	57002	40.67	38698	2.59	94
7.2	97.53	92.4	55453	102.67	100289	3.08	407
11	86.61	81.96	100998	91.27	100965	0.85	79

## Data Availability

Data are available in the Appendix A.

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
