# Peer review of "Meta-QTL and Candidate Gene Analyses of Agronomic Salt Tolerance and Related Traits in an RIL Population Derived from Solanum pimpinellifolium"

_ijms, 2024, doi:10.3390/ijms25116055_

Round 1

Reviewer 1 Report

Comments and Suggestions for Authors

Breeding for salt tolerant crops is necessary to reduce food insecurity. Prebreeding populations are fundamental materials to uncover tolerance alleles from wild germplasm. This manuscript will explain ways to exploit the genetic knowledge provided in obtaining salinity-tolerant tomato plants.Your work is very meaningful. But there are a few small shortcomings that need to be corrected.

1. The clarity of Fig. 1 is insufficient. Please provide a higher definition image.

2. Please replace the current Table 1 with a three line table.

3. Please replace the current simulation image with an actual growth image of tomatoes in Fig. 3.

4. Why haven't you located the main gene responsible for salt tolerance in tomatoes?

Comments on the Quality of English Language

The English is pretty good, I can basically understand it.

Author Response

Reviewer 1: Comments and Suggestions for Authors

Breeding for salt tolerant crops is necessary to reduce food insecurity. Prebreeding populations are fundamental materials to uncover tolerance alleles from wild germplasm. This manuscript will explain ways to exploit the genetic knowledge provided in obtaining salinity-tolerant tomato plants.Your work is very meaningful. But there are a few small shortcomings that need to be corrected.

  1. The clarity of Fig. 1 is insufficient. Please provide a higher definition image.

This figure is an image generated directly by the BioMercartor software. We cannot improve its definition.

  1. Please replace the current Table 1 with a three line table.

We have changed the location and display of this table to facilitate its reading.

  1. Please replace the current simulation image with an actual growth image of tomatoes in Fig. 3.

The aim of Fig. 3 is to represent agronomic performance (mostly based on fruit yield instead of vegetative growth) to graphically explain the strategy of rootstock utilization to improve salt tolerance of commercial cultivars in tomato (lines 272-282).

  1. Why haven't you located the main gene responsible for salt tolerance in tomatoes?

We guess you are referring to the SlHAK20 coding gene (Solyc04g008450.2) reported by Wang et al. 2020 [3]. It locates around 2.9 Mbp apart from MQTL4.1. We did not mention this in the manuscript because the experimental conditions they used were very different from field experiments. They applied only one day of salt treatment (150 mM NaCl) in a growth room.

You may also refer to the SlCBL10 coding gene (Solyc08g065330) reported by Estrada et al. 2023 [19]. It locates on chromosome 8 where we detected no productivity MQTL.

Comments on the Quality of English Language

The English is pretty good, I can basically understand it.

Reviewer 2 Report

Comments and Suggestions for Authors

The study evaluates the genetic basis of agronomic salt tolerance mechanism using meta-QTL and candidate gene analysis using a RIL population in Solanum pimpinellifolium by Maria and Emilio. Salt stress posses a sever drastic effect on the crop productivity worldwide, therefore the current study is also crucial to determining the molecular mechanism underlying salt tolerance in the crop. There are some general comments for the improvement of the current MS. There is need to revise the article from English native speaker for its readability issues specially the Introduction section.

In summary, this study could marks a substantial significance in unraveling the intricate mechanism regarding agronomic traits involved in salt tolerance in Solanum pimpinellifolium. By integrating Meta-QTL analysis with candidate gene identification, it establishes a robust framework for delving into the genetic underpinnings of multifaceted traits. This approach not only enhances our knowledge of complex genetic mechanisms but also underscores potential avenues for cultivating salt-tolerant crop varieties in the years ahead.

The authors used a comprehensive approach to use the MQTLs analysis, QTL projection, Identification of candidate genes, however, the study lacks the validation of the QTLs in the RIL population. The results section is written well, however, there are some sentences need revision for clear meaning and English editing to correct the sentence structure. Validating these QTLs genes is imperative to ascertain their robustness and elucidate their role in agronomic traits under salt stress conditions. By conducting comprehensive validation studies, we can reinforce our understanding of their significance in enhancing salt tolerance in Solanum pimpinellifolium.

Minor comments:

Unclear the meaning of “QTL across populations and cultivation conditions.” Kindly elaborate or rephrase the sentence for intended meaning.

Line 35-36: Need rephrasing to complete the sentence.

Similarly, line 46-48 needs revision for actual meaning to understand. Line 50-52; line 58-62

Line 62-64: “that will determine whether or not a gene (or a haplotype) is of agronomic”

The development of maker: or the development of marker

Line 77-80 unclear to me. Revise the sentence for intended meaning.

(Na+ , K+ , Ca2+, Cl- )  concen “back space”

those reported elsewhere [26]) I think it would be “those reported by Asins et al [26]”.

Write a suitable words for those three ways “crease in leaf water status throughout those three ways”

Write the full form at first sight and then used for abbreviation throughout the MS e.x. line 322 “gFW” “gFN”. “ LOD” Kindly check all the abbreviations used in the MS.

Line 331: elsewhere removed the word before reference “elsewhere” you already refer the citation for that specific person.

Line 336 write another suitable word “experimentwise significance level”

Comments on the Quality of English Language

There is need to revised the whole MS from native speaker or from company, for english editing and sentence structure. 

Author Response

Reviewer 2:  Comments and Suggestions for Authors

The study evaluates the genetic basis of agronomic salt tolerance mechanism using meta-QTL and candidate gene analysis using a RIL population in Solanum pimpinellifolium by Maria and Emilio. Salt stress posses a sever drastic effect on the crop productivity worldwide, therefore the current study is also crucial to determining the molecular mechanism underlying salt tolerance in the crop. There are some general comments for the improvement of the current MS. There is need to revise the article from English native speaker for its readability issues specially the Introduction section.

We have revised the manuscript readability, particularly in the introduction section. Changes are in blue.

In summary, this study could marks a substantial significance in unraveling the intricate mechanism regarding agronomic traits involved in salt tolerance in Solanum pimpinellifolium. By integrating Meta-QTL analysis with candidate gene identification, it establishes a robust framework for delving into the genetic underpinnings of multifaceted traits. This approach not only enhances our knowledge of complex genetic mechanisms but also underscores potential avenues for cultivating salt-tolerant crop varieties in the years ahead.

The authors used a comprehensive approach to use the MQTLs analysis, QTL projection, Identification of candidate genes, however, the study lacks the validation of the QTLs in the RIL population. The results section is written well, however, there are some sentences need revision for clear meaning and English editing to correct the sentence structure. Validating these QTLs genes is imperative to ascertain their robustness and elucidate their role in agronomic traits under salt stress conditions. By conducting comprehensive validation studies, we can reinforce our understanding of their significance in enhancing salt tolerance in Solanum pimpinellifolium.

Yes, we agree. As mentioned in lines 300-303, gene knockouts by CRISPR/CAS9 could be used for phenotypic validation of the candidate genes here reported. Besides, the fact that most MQTL overlap other tomato yield QTL that were previously detected using various populations, cultivation conditions, and statistical methodologies (Table 2) suggests they are quite stable.

Minor comments:

Unclear the meaning of “QTL across populations and cultivation conditions.” Kindly elaborate or rephrase the sentence for intended meaning.

Ok. This sentence has been rephrased in the abstract and conclusion sections (Lines 14-15, 372-373). It is explained in the first paragraph of section 3.3.

Line 35-36: Need rephrasing to complete the sentence.

It has been rephrased (lines 34-36).

Similarly, line 46-48 needs revision for actual meaning to understand. Line 50-52; line 58-62

All of them have been changed.

Line 62-64: “that will determine whether or not a gene (or a haplotype) is of agronomic”

We have substituted haplotype by “a given set of linked genes” (line 54).

The development of maker: or the development of marker

Yer, marker.

Line 77-80 unclear to me. Revise the sentence for intended meaning.

The sentence has been completed (line 81).

(Na+ , K+ , Ca2+, Cl- )  concen “back space”

Ok.

those reported elsewhere [26]) I think it would be “those reported by Asins et al [26]”.

Ok.

Write a suitable words for those three ways “crease in leaf water status throughout those three ways”

It has been rephrased (lines 203-206).

Write the full form at first sight and then used for abbreviation throughout the MS e.x. line 322 “gFW” “gFN”. “ LOD” Kindly check all the abbreviations used in the MS.

LOD and other abbreviations have been written in the full form at first sight. The “g” prefix used in “gFW” and “gFN” is explained in lines 320-321 and Table S4.

Line 331: elsewhere removed the word before reference “elsewhere” you already refer the citation for that specific person.

Ok.

Line 336 write another suitable word “experimentwise significance level”

We cannot change it; this is a statistical term in a test involving multiple comparisons. It is defined as the probability of making at least one Type I error over an entire research study.

Comments on the Quality of English Language

There is need to revised the whole MS from native speaker or from company, for english editing and sentence structure.

We have tried to improve it.

Reviewer 3 Report

Comments and Suggestions for Authors

The objectives of the manuscript were the QTL re-analysis of previously evaluated fruit yield and reproductive traits under salt conditions in a RIL population derived from S. pimpinellifolium by using a SNP-saturated linkage map, and to identify putative candidate genes. QTL for reproductive traits and several QTL for fruit yield components not detected in previous studies were reported. The clustering of QTL was investigated by meta-QTL analysis. Sixty QTL were grouped into 12 productivity MQTL. A list of putative candidate genes was reported for each MQTL and discussed concerning the importance of some physiological traits contributing to salt tolerance.

The title adequately describes the subject of the manuscript and the abstract briefly tells what was done and summarizes the main results and conclusions. Materials, experimental design, and methods are appropriate and adequately described.  The conclusions are adequate and supported by the data. The information is presented in a relatively simple manner to be understood by a competent scientist or reader. The subject falls within the scope of the “International Journal of Molecular Sciences”.

-      -  Check the use of commas and full stops in Figures 1 and S1, and Table 1, S1, and S5 (the standard English punctuation for commas and full stops should be followed).

-     -   Check the Latin name of scientific plant species in the text: it should be always in italics.

Author Response

Reviewer 3: Comments and Suggestions for Authors

The objectives of the manuscript were the QTL re-analysis of previously evaluated fruit yield and reproductive traits under salt conditions in a RIL population derived from S. pimpinellifolium by using a SNP-saturated linkage map, and to identify putative candidate genes. QTL for reproductive traits and several QTL for fruit yield components not detected in previous studies were reported. The clustering of QTL was investigated by meta-QTL analysis. Sixty QTL were grouped into 12 productivity MQTL. A list of putative candidate genes was reported for each MQTL and discussed concerning the importance of some physiological traits contributing to salt tolerance.

The title adequately describes the subject of the manuscript and the abstract briefly tells what was done and summarizes the main results and conclusions. Materials, experimental design, and methods are appropriate and adequately described.  The conclusions are adequate and supported by the data. The information is presented in a relatively simple manner to be understood by a competent scientist or reader. The subject falls within the scope of the “International Journal of Molecular Sciences”.

-      -  Check the use of commas and full stops in Figures 1 and S1, and Table 1, S1, and S5 (the standard English punctuation for commas and full stops should be followed).

Ok. It has been done.

-     -   Check the Latin name of scientific plant species in the text: it should be always in italics.

Ok. It has been done.

Round 2

Reviewer 2 Report

Comments and Suggestions for Authors

The authors addressed the comments carefully to improve the article, making it suitable for publication in IJSM.